# Scrambler Therapy for Chronic Pain after Burns and Its Effect on the Cerebral Pain Network: A Prospective, Double-Blinded, Randomized Controlled Trial

**DOI:** 10.3390/jcm11154255

**Published:** 2022-07-22

**Authors:** Seung Yeol Lee, Chang-hyun Park, Yoon Soo Cho, Laurie Kim, Ji Won Yoo, So Young Joo, Cheong Hoon Seo

**Affiliations:** 1Department of Physical Medicine and Rehabilitation, Soonchunhyang University Bucheon Hospital, College of Medicine, Soonchunhyang University, Bucheon 14158, Korea; shouletz@gmail.com; 2Center for Neuroprosthetics and Brain Mind Institute, Swiss Federal Institute of Technology (EPFL), 1202 Geneva, Switzerland; park.changhyun@hanmail.net; 3Department of Rehabilitation Medicine, Hangang Sacred Heart Hospital, College of Medicine, Hallym University, Seoul 07247, Korea; hamays@daum.net; 4Kirk Kerkorian School of Medicine, University of Nevada Las Vegas, Las Vegas, NV 89102, USA; kiml17@unlv.nevada.edu; 5Department of Internal Medicine, School of Medicine, University of Nevada Las Vegas, Las Vegas, NV 89102, USA; ji.yoo@unlv.edu

**Keywords:** cerebral pain network, scrambler therapy, burn, chronic pain

## Abstract

Chronic pain is common after burn injuries, and post-burn neuropathic pain is the most important complication that is difficult to treat. Scrambler therapy (ST) is a non-invasive modality that uses patient-specific electrocutaneous nerve stimulation and is an effective treatment for many chronic pain disorders. This study used magnetic resonance imaging (MRI) to evaluate the pain network-related mechanisms that underlie the clinical effect of ST in patients with chronic burn-related pain. This prospective, double-blinded, randomized controlled trial (ClinicalTrials.gov: NCT03865693) enrolled 43 patients who were experiencing chronic neuropathic pain after unilateral burn injuries. The patients had moderate or greater chronic pain (a visual analogue scale (VAS) score of ≥5), despite treatment using gabapentin and other physical modalities, and were randomized 1:1 to receive real or sham ST sessions. The ST was performed using the MC5-A Calmare device for ten 45 min sessions (Monday to Friday for 2 weeks). Baseline and post-treatment parameters were evaluated subjectively using the VAS score for pain and the Hamilton Depression Rating Scale; MRI was performed to identify objective central nervous system changes by measuring the cerebral blood volume (CBV). After 10 ST sessions (two weeks), the treatment group exhibited a significant reduction in pain relative to the sham group. Furthermore, relative to the pre-ST findings, the post-ST MRI evaluations revealed significantly decreased CBV in the orbito-frontal gyrus, middle frontal gyrus, superior frontal gyrus, and gyrus rectus. In addition, the CBV was increased in the precentral gyrus and postcentral gyrus of the hemisphere associated with the burned limb in the ST group, as compared with the CBV of the sham group. Thus, a clinical effect from ST on burn pain was observed after 2 weeks, and a potential mechanism for the treatment effect was identified. These findings suggest that ST may be an alternative strategy for managing chronic pain in burn patients.

## 1. Introduction

Up to one-third of patients suffer from chronic pain after burns [1,2]. Management of chronic pain after burn injuries is critical because it affects a patient’s quality of life [3]. There are diverse treatments for chronic pain after burn injury; however, due to the incompleteness of information regarding the underlying causes, most burn centers use new treatment approaches without standard treatment protocols.

Chronic pain in burn patients can be treated using pharmacological strategies (e.g., NSAIDS, opioids, tramadol, and anticonvulsants) and non-pharmacological modalities (e.g., compression garments, physical therapy, and occupational therapy) [4,5]. Many reports have described the clinical effects of peripheral nerve electrostimulation for chronic pain [6,7]. Scrambler therapy (ST) is a non-invasive electrostimulation treatment that blocks pain, and recently it has been proven effective for treating chronic pain syndromes [8]. This mechanism involves sending synthetized “non-pain” information that looks like depolarization currents through the C-fibers of afferent pathways to the relevant brain center(s), leading to remodulation of the pain network [9,10,11]. Clinical effects related to neurons other than C-fibers have also been demonstrated, such as numbness or tingling sensations, and there is increasing interest in clarifying the therapeutic mechanism of ST [8]. 

It has been suggested that burn injuries trigger the peripheral nerves and cause sensitization of the nociceptive fibers [12,13]; chronic pain in these cases is explained mostly by the inflammatory response at hypertrophic scars [14]. However, chronic pain in this setting is also related to a mechanism that cannot be explained by peripheral causes alone; recent reports indicate that changes in the cerebral pain network occur in cases of musculoskeletal diseases with chronic pain [15,16,17]. Brain imaging studies have been conducted on the mechanisms of chronic pain aggravation and pain improvement after pain treatment [16,18,19,20]. The cerebral blood volume (CBV) is a hemodynamic variable that represents the fraction of the cerebral tissue volume occupied by blood at a given time point, which is highly correlated with oxygen metabolism. Although lower in temporal resolution than deoxyhemoglobin fMRI, T1-weighted post gadolinium measurement of CBV offer resolution superior to other techniques and is the only fMRI measure that is currently amenable to submillimeter resolution functional maps [21]. This randomized controlled trial was to investigate the pain-suppressing effect of ST and the mechanism of its effect on the cerebral pain network in patients with chronic neuropathic pain caused by burns. 

## 2. Methods

This prospective, double-blind, randomized controlled trial enrolled 36 male and 7 female patients at the Department of Rehabilitation Medicine at Hangang Sacred Heart Hospital in Korea between March 2018 and October 2020. All patients provided written informed consent. The trial protocol was registered at ClinicalTrials.gov (NCT03865693) and was approved by the ethics committee of Hangang Sacred Heart Hospital (HG2017-088). This study was conducted according to the CONSORT guidelines.

### 2.1. Clinical Subjects

To be included in the study, patients had to (i) be 18 years of age or older; (ii) have burn scars that had re-epithelialized after aseptic care or skin grafting; (iii) have complained of moderate or greater chronic neuropathic pain with reduced sensation in the burned area (≥5 points on a 10-point visual analog scale (VAS)) lasting for >3 months after the burn injury, despite receiving pharmaceutical treatment and/or physical therapy [22,23]; (iv) a burn on either right or left side of the body; and (v) be dominant in the right hand. Patients were excluded based on the following criteria: (i) a history of cardiac arrest; (ii) neurological disease, or brain surgery; (iii) unstable heart disease or presence of a cardiac pacemaker; (iv) pain resulting from other causes (e.g., neuromuscular diseases) as confirmed via imaging (radiography, ultrasonography, computed tomography, or MRI); (v) psychiatric disorders; (vi) abnormal renal function; (vii) contraindication for MRI; (viii) pregnancy; or (ix) a score of 8 or higher on the Korean version of the Hamilton depression rating scale (HDRS) [24]. Possible drug effects were minimized by excluding patients who were receiving extended-release morphine therapy. Antiepileptic and antidepressant drugs, which are pain medications that affect brain activity, were maintained at their dosage throughout the study [25]. Numbers were assigned according to the order of admission to 43 burn patients who satisfied all the study criteria. A computer program was used to divide the patients randomly into either the ST group (n = 20) or sham group (n = 23). 

The study used an MC-5A Pain Scrambler Therapy device (Competitive Technologies, Inc., Fairfield, CT, USA) with electrode patches applied to a 20–25 mm area surrounding the site of the most painful burn scar. In the ST group, the stimulus intensity was treated as the maximum intensity that the patient could tolerate without discomfort (≤70 U) (Figure 1). In the sham group, the stimulation was maintained at a non-therapeutic threshold (0–10 U for the duration of the session) [26,27]. Both the ST and sham groups completed a total of 10 sessions (Monday to Friday over 2 weeks), and each session lasted 45 min. 

### 2.2. Clinical Assessments

Only the therapist had access to the allocation schedule, and all participants and evaluators were blinded to the treatment details. Depressive mood was assessed using the Korean version of the Hamilton Depression Rating Scale (HDRS) [24]. Patients were asked to score their mean pain intensity using the VAS and Brief Pain Intensity (BPI). The VAS uses a 10-point scale with responses ranging from 0 (no pain) to 10 (worst pain ever experienced). The BPI consists of a sensory dimension that indicates the intensity of pain and a reactive dimension that measures the decrease in function due to pain [28]. The VAS tool and MRI scan were administered at baseline and after 2 weeks of treatment; it primarily aimed to identify brain network changes and the relationship between any brain network changes and the intensity of pain. 

### 2.3. MRI Acquisition and CBV Mapping

All MRIs were performed using a 3.0 T magnetic resonance scanner (MAGNETOM Skyra, Siemens; Erlangen, Germany) and an established steady-state gadolinium-enhanced MRI technique [18,29]. Two high-resolution T1-weighted images were acquired for each participant: one before a standard intravenous administration of gadolinium contrast agent (0.1 mmol/kg gadoterate meglumine) and the second at 4 min after the gadolinium administration. T2-weighted and diffusion-weighted MRI images were taken to exclude patients with suspected parenchymal injury [18]. A single-shot diffusion-weighted echo planar imaging sequence was taken in the same manner as Joo et al. to assess the CBV changes caused by the burn-related chronic pain [18]. Processing of the CBV mapping data was performed using SPM12 software (https://www.fil.ion.ucl.ac.uk/spm/software/spm12/, accessed on 8 May 2022), as previously described [21,30]. The pre- and post-contrast images were spatially transformed to the same standard space, and then a map of the contrast-induced signal difference ratios was acquired using the equation: (post-contrast signal − pre-contrast signal)/(maximum signal difference in the superior sagittal sinus) × 100. For patients with injuries on the left extremities, the map was rotated around the midsagittal axis. The CBV maps were analyzed using a randomization tool [31] that implements nonparametric permutation inference on neuroimaging data. For each map, voxel-wise comparisons between groups and voxel-wise correlations with VAS scores were evaluated after adjusting for each individual’s age, sex, and degree of depression. The CBV valuations were performed at baseline and after 2 weeks of ST or sham treatment. 

### 2.4. Statistical Analysis

All statistical analyses were performed using SPSS software (version 23.0; IBM Corp., Armonk, NY, USA). Non-parametric measurements between the two groups were analyzed using the Mann–Whitney test. After testing for data normality, Mann–Whitney U tests were performed to compare pretreatment homogeneity between the groups for total body surface area, days between the burn injury and MRI acquisition, VAS score, BPI score, and HDRS score. Fisher’s exact test was used to assess differences in the sexes and the sites of burn injury between the groups.

Pain scores from before and after treatment within each group were compared using the Wilcoxon signed rank sum test. The comparison between the VAS scores of the two groups after 2 weeks of treatment was performed using the Mann–Whitney test. After correction for multiple comparisons with threshold-free cluster enhancement, group differences and VAS correlations with a *p*-value < 0.05 were considered statistically significant [32]. 

## 3. Results

### 3.1. Clinical Features

Six out of 20 patients in the ST group did not need further pain treatment due to pain improvement; therefore, follow-up MRI scans could not be performed after 2 weeks due to lack of patients participation. In the final enrollment, there were 14 participants in the ST group and 23 in the sham group. No significant differences in age, sex, or other baseline characteristics were observed between the groups (Table 1). There was no significant difference between the two groups for the baseline pain scores or CBV (*p* > 0.05). Relative to the baseline value, the ST group showed a significant reduction in the pain score after ST (*p* = 0.004); the sham group also had a significant reduction in the pain score after 2 weeks of therapy (*p* = 0.001). After 2 weeks of treatment, it was found that the VAS scores of the ST group were significantly decreased compared with the VAS scores of the sham group (*p* < 0.001) (Table 2). 

### 3.2. CBV Mapping before and after Scrambler Therapy

There was no difference in baseline CBV before ST between the two groups (*p* > 0.05). In the ST group, the CBVs of the right orbito-frontal gyrus (*p* = 0.004), right middle frontal gyrus (*p* = 0.004), right superior frontal gyrus (*p* = 0.004), right gyrus rectus (*p* = 0.004), left orbito-frontal gyrus (*p* = 0.004), and left superior frontal gyrus (*p* = 0.004) were decreased after 2 weeks of ST (Figure 2 and Table 3).

After 2 weeks of treatment, it was observed that the CBV had increased in the precentral gyrus (*p* = 0.002) and postcentral gyrus (*p* = 0.004) of the hemisphere associated with the burned limb in the ST group, as compared with the CBV of the sham group (Figure 3 and Table 4). No serious side effects occurred that were potentially related to ST.

## 4. Discussion

This study aimed to evaluate the clinical usefulness of ST and identify the pain network alterations associated with ST for chronic neuropathic pain in burn patients. The ST group experienced pain reduction from ST and a hypersensitization mechanism potentially related to treatment was observed. These results suggest that ST may demonstrate an alternative strategy to manage chronic pain in burn patients.

Pain perception is affected by alterations in the cerebral pain network [33]; chronic neuropathic pain is caused by hyperexcitability. Studies have also identified an important role for central plasticity, given the long-term nature of chronic pain [3,34,35]. The various dimensions of pain intensity are associated with different areas of the pain network. Changes in the pain and motor network have been observed when chronic pain occurs in burn patients [29]. The treatment mechanism of ST is known to inhibit pain by stimulating C fibers of afferent pathways with five artificial electrical stimulation and blocking the pathways of pain information [36]. Several studies have indicated that ST provides clinical benefits in cases of chronic and neuropathic pain, with a pain reduction effect that can last up to 1 year after 2 weeks of therapy [9,36]. The long-term effect is thought to involve reduced central sensitization, due to the delivery of a signal to the surface receptors of the C-fibers, which transmit “non-pain” information along the damaged pathways [10]. Joo et al. evaluated symptom improvement after ST for severe pruritus, which has a similar mechanism to neuropathic pain in burn patients, and found that ST was safe [21]. The maintenance of pain reduction after ST is explained as a central pain processing mechanism related to memory for pain [27]. 

We collected CBV maps from before ST and after 10 therapy sessions. After 2 weeks of ST, a decrease was observed in the CBVs of the bilateral orbito-frontal gyrus, right middle frontal gyrus, bilateral superior frontal gyrus, and right gyrus rectus in the ST group. In addition, the CBV was increased in the precentral gyrus and postcentral gyrus of the hemisphere associated with the burned limb in the ST group, as compared with that of the sham group. Burn patients may suffer from chronic pain, which is characterized by neuroanatomical plasticity, including central sensitization. Many studies of non-invasive brain-modulatory therapy have demonstrated a relationship between changes in the cerebral pain network and pain intensity. This pain network consists of the primary and secondary somatosensory cortex, the prefrontal lobe, dorsolateral prefrontal cortex, and medial prefrontal cortex [29]. Neuroimaging studies indicate that the dorsolateral prefrontal cortex may play a role in a top-down mode of inhibition via fibers descending from the prefrontal cortex, which may modulate pain perception. Inhibiting the activity of the dorsolateral prefrontal cortex reduces pain by removing transcallosal inhibition and allowing an enhanced descending inhibition from the contralateral hemisphere [20]. Portilla et al. reported that non-invasive brain stimulation over the precentral gyrus causes decreased cortical excitability that cause chronic neuropathic pain in the occipital and frontal areas [37]. Recently, non-invasive brain stimulation has been applied to burn patients as a new treatment method for pain and neuropathy that show no improvement from other conventional therapies. Stimulation of the precentral gyrus results in compensating sensory nerve loss due to burns [28]. The mechanism of central sensitization in patients with chronic musculoskeletal pain is explained by a decreased activation in the postcentral gyrus [38,39,40]. Hosseini et al. reported that modulation of the sensory cortex with transcranial direct current stimulation reduced pain and anxiety during burn scar treatment [41]. A review article also indicated that changes in cerebral activation during non-invasive brain stimulation were associated with improvements in pain symptoms, pruritus, depression, and sleep disturbances caused by burns [42]. 

This study explored the effects of ST based on subjective and objective parameters, including cerebral activity in patients with chronic neuropathic pain after burn injury. This study had some limitations, including a small sample size and a short follow-up period. Due to the small sample size, the effects of individual differences in the side of the injuries could not be fully taken into account. The majority of patients with severe burns admitted to this burn center, and consequently enrolled in this study, were male. Thus, it would be inappropriate to generalize the study results to both sexes. For ethical reasons, the patients’ pharmacological and non-pharmacological treatments were maintained the same as before the start of the study; therefore, further study that excludes these aspects is needed. Further studies are needed to develop advanced strategies for controlling neuropathic pain caused by thermal injury, including cranial magnetic stimulation or transcranial direct current stimulation. Future research should also assess the effects of ST on central neurotransmitters, which play a major role in the endogenous pain facilitatory and inhibitory pathways. In order to confirm the therapeutic effect, it is thought that more research is needed on clinical parameters, including not only the pain level, but also sleep disturbances and changes in daily living activities due to pain.

## 5. Conclusions

This study found that burn patients with chronic neuropathic pain had CBV changes in regions of the cerebral pain network that are associated with the frontal lobe, precentral gyrus, and postcentral gyrus. Thus, a clinical effect from ST on burn pain was observed after 2 weeks, and a potential mechanism for the treatment effect was identified. These findings suggest that ST may be an alternative strategy for managing chronic pain in burn patients.

## Figures and Tables

**Figure 1 jcm-11-04255-f001:**
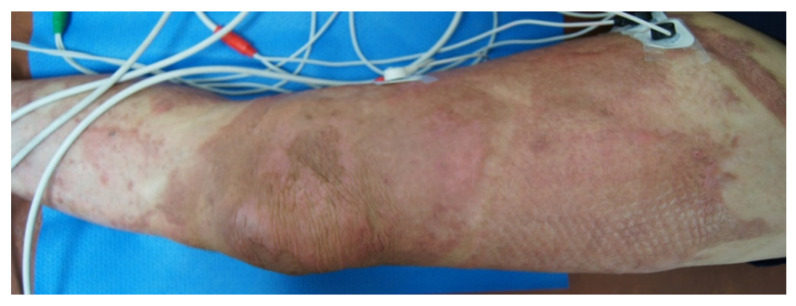
Pain scrambler therapy on scar pain site in burn patients.

**Figure 2 jcm-11-04255-f002:**
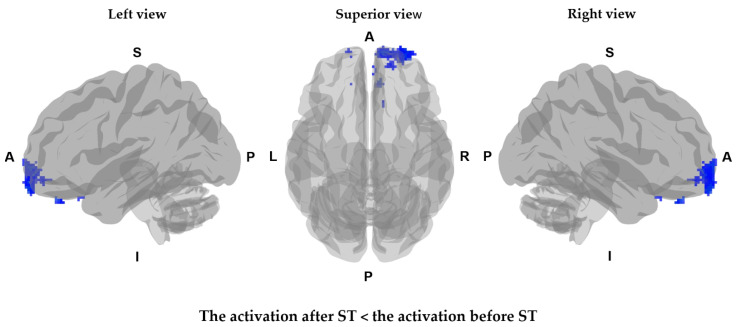
Mapping of CBV status after ST versus CBV status before ST as adjusted for sex and age. The brain regions marked in blue are regions with decreased activation after ST treatment in the ST group. A, anterior; P, posterior; L, left; R, right; I, Inferior; S, Superior.

**Figure 3 jcm-11-04255-f003:**
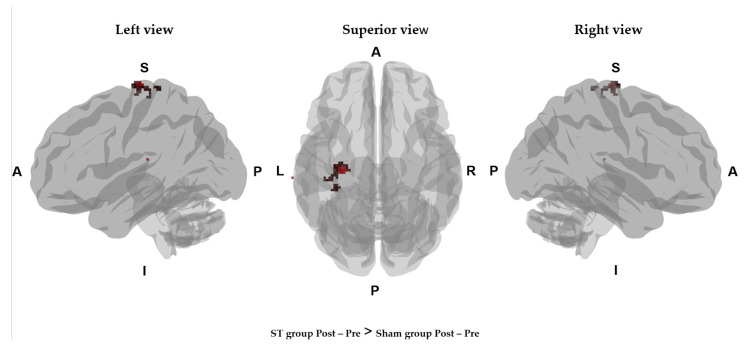
Mapping of the CBV states in the experimental group versus the sham group. The brain regions (the precentral gyrus and postcentral gyrus) marked in red are regions with increased activation after treatment in the ST group compared to the Sham group. A, anterior; P, posterior; L, left; R, right.

**Table 1 jcm-11-04255-t001:** Demographic data of subjects.

	Experimental Group (n = 14)	Sham Group (n = 23)	*p*
Male:Female	12:2	18:5	0.69
Age (years)	47 (39–59)	49 (30–57)	0.83
TBSA (%)	11 (5–23)	20 (5–23)	0.65
Days between burn and MRI acquisition	76 (53–93)	74 (52–107)	1.00
The sites of burn injury			
Arm: Forearm/Hand: Thigh: Leg/Foot	5:5:1:3	6:8:1:8	0.82
VAS	6 (5–8)	7 (6–8)	0.17
BPI			
Sensory dimension	23 (13–26)	26 (18–30)	0.29
Reactive dimension	38 (27–47)	42 (34–50)	0.41
HDRS	4 (1–4)	2 (1–4)	0.44

TBSA, total burn surface area; VAS, visual analog scale; HDRS, Hamilton depression rating scale; Values are presented as median (IQR), *p*-values were calculated using Fisher’s exact test or the Mann–Whitney test.

**Table 2 jcm-11-04255-t002:** Comparison of VAS score before and after treatment between two groups.

	Experimental Group			Sham Stimulation Group		
	Baseline	After 2 Weeks	*p*	Baseline	After 2 Weeks	*p*
VAS, median (IQR)	6 (5–8)	3 (3–4)	0.004	7 (6–8)	6 (5–7)	0.001
Comparison of VAS after 2 weeks between groups	<0.001					

VAS, visual analog scale.

**Table 3 jcm-11-04255-t003:** Clusters of decreased CBV after scrambler therapy relative to CBV before scrambler therapy.

Comparison	Cluster No	Voxel Count	Grey Matter Label	*T* Value	*p* Value	Coordinates (mm)		
						*x*	*y*	*z*
Decreased CBV	1	232	Right orbito-frontal gyrus	450.661	0.004	30	60	−8
			Right middle frontal gyrus	441.674	0.004	26	58	−6
			Right superior frontal gyrus	337.098	0.004	26	66	−6
	3	16	Right gyrus rectus	252.559	0.004	8	38	−30
	4	5	Left orbito-frontal gyrus	356.164	0.004	−16	64	−12
			Left superior frontal gyrus	345.580	0.004	−14	64	−12

CBV, cerebral blood volume.

**Table 4 jcm-11-04255-t004:** Comparisons CBV clusters between the experimental and sham groups.

Comparison	Gray Matter Label	*T* Value	*p* Value	Coordinates (mm)		
				*x*	*y*	*z*
ST_post – pre_ >Sham _post − pre_	Left precentral gyrus	411.954	0.002	−30	−20	74
	Left postcentral gyrus	373.215	0.004	−32	−26	72

CBV, cerebral blood volume.

## Data Availability

The data presented in this study are available on request from the corresponding author. The data are not publicly available due to the participants’ sensitive personal information.

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
