# Peer review of "Scrambler Therapy for Chronic Pain after Burns and Its Effect on the Cerebral Pain Network: A Prospective, Double-Blinded, Randomized Controlled Trial"

_jcm, 2022, doi:10.3390/jcm11154255_

Round 1
Reviewer 1 Report
Authors performed a prospective, randomized, sham-controlled trial of patients with severe chronic pain (≥5 points on a 10-point visual analog scale [VAS]) lasting for >3 months after the burn injury. The objective was to assess Scrambler therapy for reducing pain. The ST was performed using the MC5-A Calmare® device (Competitive Technologies, Inc. Fairfield, USA) for ten 45-min sessions (Monday to Friday for 2 weeks). The study enrolled 14 ST and 23 sham patients looking at 2 week outcomes. The authors concluded the clinical effects of ST treatment on severe chronic pain in burn patients and the mechanism of the clinical effect could be confirmed. I have the following concerns regarding limitations of the study. These include the following which should be addressed within the manuscript:
-2 week pain assessment.
-Small sample size (14 vs. 23). High risk of type 1 error.
-No power analysis. High risk of type 2 error.
-Was the data normal or parametric? No mention of sample distribution. This is difficult to believe with the sample size and no analysis. This may also be the reason why certain findings were significant.
-Male:female ratio in groups were disproportionate (Experimental 12:2 vs Sham 18:5)
-Non-standardized definitions of severe chronic pain. >3 months
-No long-term patient follow-up
Lines 46-47: “The clinical effect of ST treatment on chronic pain in burn patients and the mechanism of the clinical effect could be confirmed.”
Authors should soften the language of the conclusions and include “at 2 week follow-up.” “We observed a clinical effect of ST treatment on pain in burn patients and the potential mechanism of the clinical effect at 2 week follow-up.”
Lines 57-58: “Chronic pain in burn patients can be treated using pharmacological strategies (anticonvulsants and antidepressants) and non-pharmacologic modalities
The list of pharmacologic agents and non-pharmacologic modalities is well beyond these classes. There are medications like NSAIDs, acetaminophen, opioids, ascorbic acid, tramadol, lidocaine, capsaicin. Non-pharmacologic modalities like lasers, fat grafting, nerve repair, nerve decompression, occupational therapy, physical therapy, compression garments.
Lines 61-63: “This mechanism involves sending synthetized “non-pain” information, which looks like depolarization currents, through the C-fibers to the relevant brain center(s) and leading to remodulation of the pain network [9-11].”
Does this target afferent, efferent or both pathways?
Line 78: “enrolled 36 male patients and 7 female patients”
This should be addressed as a limitation in the discussion. Male and Female sexes were not evenly distributed.
Methods
85-86: “had complained of severe chronic pain (≥5 points on a 10-point visual analog scale [VAS]) lasting for >3 months after the burn injury”
Where did this definition originate from? Is a value of 5 considered the standardized definition of severe? Can the authors provide the reference? Many would not consider something as chronic unless it was >6 months
95-96: “Eligible participants were assigned identification numbers in the order of admission and randomly allocated to a ST group (20 patients) or a sham group (23 patients).”
How were these patients randomized? Were ST group patients given even numbers or odd numbers? It would be worth mentioning the randomization process in the main manuscript. This will help readers to determine different risks of bias. Authors should also briefly describe the blinding process. How was the ST dose administered at a subtherapeutic threshold without the provider and patient knowing? Who administered the treatment?
Line 113: “after 2 weeks treatment”
This is a major limitation of the study. The authors will need to demonstrate these network changes are maximized at 2 weeks. Otherwise, I would not expect there to be a difference in chronic pain networks in the short interval of 2 weeks. If these patients have chronic pain, a minimum follow-up should at least be 6 months.
141: “P-value of <0.05”
Was this two-tailed?
164: “Table 2. Comparison of VAS score before and after treatment between two groups”
Line 178: “Table 3. Clusters of decreased CBV after scrambler therapy relative to CBV before scrambler therapy”
Are the p-values really 0.004 for all these results? This seems strange.
Line 206: “Discussion”
Authors need to include a limitations section.
-2 week pain assessment.
-Small sample size (14 vs. 23). High risk of type 1 error.
-No power analysis. High risk of type 2 error.
-Was the data normal or parametric? No mention of sample distribution. This is difficult to believe with the sample size and no analysis. This may also be the reason why certain findings were significant.
-Male:female ratio in groups were disproportionate (Experimental 12:2 vs Sham 18:5)
-Non-standardized definitions of severe chronic pain. >3 months
-No long-term patient follow-up
Can the authors address these limitations?
Lines 257-259: “This study proved that central sensitization was suppressed using ST in burn patients with chronic pain, and the clinical effect along with the treatment mechanism was confirmed.”
This statement overestimates the results of the study. Would need to include in the conclusion for patients with pain >3 months assessed at 2 weeks follow-up.
Author Response
- I have the following concerns regarding limitations of the study. These include the following which should be addressed within the manuscript:-2 week pain assessment;-Small sample size (14 vs. 23). High risk of type 1 error; No power analysis. High risk of type 2 error-Was the data normal or parametric? No mention of sample distribution. This is difficult to believe with the sample size and no analysis. This may also be the reason why certain findings were significant.-Male:female ratio in groups were disproportionate (Experimental 12:2 vs Sham 18:5)-Non-standardized definitions of severe chronic pain. >3 months-No long-term patient follow-up
Answer> We agree with your opinions. For regarding the contents you pointed out, we have added limitations and additional research to be conducted in the future to the discussion section.
- Lines 46-47: “The clinical effect of ST treatment on chronic pain in burn patients and the mechanism of the clinical effect could be confirmed.” ; Authors should soften the language of the conclusions and include “at 2 week follow-up.” “We observed a clinical effect of ST treatment on pain in burn patients and the potential mechanism of the clinical effect at 2 week follow-up.”
Answer> We appreciate you careful advise. As you pointed out, I have corrected the sentence faithfully to the results and not to exaggerate. We hope that these modifications will be better understood by the reader.
- Lines 57-58: “Chronic pain in burn patients can be treated using pharmacological strategies (anticonvulsants and antidepressants) and non-pharmacologic modalities.; The list of pharmacologic agents and non-pharmacologic modalities is well beyond these classes. There are medications like NSAIDs, acetaminophen, opioids, ascorbic acid, tramadol, lidocaine, capsaicin. Non-pharmacologic modalities like lasers, fat grafting, nerve repair, nerve decompression, occupational therapy, physical therapy, compression garments.
Answer> We appreciate you careful advise. I totally agree with reviewer’s comment. By focusing only on this study, the example for pharmacologic and nonpharmacologic treatment was modified.
- Lines 61-63: “This mechanism involves sending synthetized “non-pain” information, which looks like depolarization currents, through the C-fibers to the relevant brain center(s) and leading to remodulation of the pain network [9-11].” Does this target afferent, efferent or both pathways?
Answer> We appreciate you careful advise. The authors added that they relate to the afferent pathway. We hope that these modifications will be better understood by the reader.
- Line 78: “enrolled 36 male patients and 7 female patients”; This should be addressed as a limitation in the discussion. Male and Female sexes were not evenly distributed.
Answer> We appreciate you careful advise. The patient group of the burn center is the majority of patients due to accidents on the job, and male is the majority of the patient group, so a limitation has been added to the discussion section.
- 85-86: “had complained of severe chronic pain (≥5 points on a 10-point visual analog scale [VAS]) lasting for >3 months after the burn injury”
Where did this definition originate from? Is a value of 5 considered the standardized definition of severe? Can the authors provide the reference? Many would not consider something as chronic unless it was >6 months
Answer> We appreciate you careful advise. As you pointed out, the pain score of 5 or more was corrected to moderate or more pain, and reference were added. We added a reference for chronic pain over 3 months.
Dydyk AM, Grandhe S. Pain Assessment. StatPearls. Treasure Island (FL): StatPearls PublishingCopyright © 2022, StatPearls Publishing LLC.; 2022.
Jensen MP, Chen C, Brugger AM. Interpretation of visual analog scale ratings and change scores: a reanalysis of two clinical trials of postoperative pain. The journal of pain. 2003;4:407-14.
- 95-96: “Eligible participants were assigned identification numbers in the order of admission and randomly allocated to a ST group (20 patients) or a sham group (23 patients).” How were these patients randomized? Were ST group patients given even numbers or odd numbers? It would be worth mentioning the randomization process in the main manuscript. This will help readers to determine different risks of bias. Authors should also briefly describe the blinding process. How was the ST dose administered at a subtherapeutic threshold without the provider and patient knowing? Who administered the treatment?
Answer> We appreciate you careful advise. The contents of the randomization method have been modified.
- Line 113: “after 2 weeks treatment”; This is a major limitation of the study. The authors will need to demonstrate these network changes are maximized at 2 weeks. Otherwise, I would not expect there to be a difference in chronic pain networks in the short interval of 2 weeks. If these patients have chronic pain, a minimum follow-up should at least be 6 months.
Answer> We appreciate you careful advise. The purpose of this study was to prove the effect of non-pharmacologic treatment in addition to pharmacologic treatment and to observe the treatment mechanism in patients with chronic pain after burns. The study design was conducted with a 2-week follow-up. The need for further study and limitations have been added to the discussion section.
- 141: “P-value of <0.05”, Was this two-tailed?
Answer> We appreciate the reviewer's comment. All statistical inferences for CBV maps were conducted by using one-tailed tests at the specified significance level. In SPM12 software that we used for the processing and statistical analysis of CBV maps, one-tailed tests are applied by default because a hypothesized relationship regarding imaging usually needs to be specified by its direction (larger than or smaller than; or positive or negative) in describing the mechanisms underlying behaviors.
- 164: “Table 2. Comparison of VAS score before and after treatment between two groups”
Answer> We appreciate you careful advise. Table 2 compares the pain index before and after treatment within each group, and also compares the pain index after 2 weeks between the two groups.
- Line 178: “Table 3. Clusters of decreased CBV after scrambler therapy relative to CBV before scrambler therapy”; Are the p-values really 0.004 for all these results? This seems strange.
Answer> We appreciate you careful advise. It was confirmed that the P-values are the same even after analyzing again.
- Authors need to include a limitations section in the discussion section.
-2 week pain assessment,-Small sample size (14 vs. 23). High risk of type 1 error,-No power analysis. High risk of type 2 error,-Was the data normal or parametric? No mention of sample distribution. This is difficult to believe with the sample size and no analysis. This may also be the reason why certain findings were significant,-Male:female ratio in groups were disproportionate (Experimental 12:2 vs Sham 18:5),-Non-standardized definitions of severe chronic pain. >3 months,-No long-term patient follow-up, Can the authors address these limitations?
Answer> We agree with your opinions. For regarding the contents you pointed out, we have added limitations and additional research to be conducted in the future to the discussion section.
- Lines 257-259: “This study proved that central sensitization was suppressed using ST in burn patients with chronic pain, and the clinical effect along with the treatment mechanism was confirmed.”; This statement overestimates the results of the study. Would need to include in the conclusion for patients with pain >3 months assessed at 2 weeks follow-up.
Answer> We appreciate you careful advise. The sentence in the conclusion part has been corrected. We hope that these modifications will be better understood by the reader.

Reviewer 2 Report
The paper could be of interest provided the authors report more precisely methods, results and data interpretation.
Line 35: ….d severe chronic pain (a visual analogue scale [VAS] score of ≥5)… usually severe paini s for VAS >7/10; perhaps authors should indicate moderate-to-severe pain or moderate or severe pain; however, the most important issue is how disabling was the pain.
Line 71: I am not sure that most readers are familiar with CBV techniques. The authors should spend 1 or 2 paragraphs explaining what it does actually measure and its advantages/disavantages compare to rsMRI and fMRI (Ke Zhang et al., 2018)
Line 128: for patients with injuries on the left extremities, the map was flipped around the mid-sagittal axis…however, cortical non sensorimotor areas in the left and right hemisphere are not equivalent (Manss, 2019)...were patients all right-handed?
Line 159: HRDS, Hamilton depression rating scale, scores are very low for patients with chronic intense pain…they do not seem real… in any event did depression scores changed after treatment?
Line 164: sham stimulation did not cause any change in pain VAS… however, placebo effect usually accounts for 20 to 60% of the analgesic effect…see Thibaut et al., 2019 (within authors references)
Line 168: …There was no difference in baseline CBV before ST between two groups…There should have been some (Joo et al., 2021)… why they did not find any?
Which was the effect of the sham stimulation or, if you will, the placebo stimulation?
Line 203: ST increase CBV in the left upper pre- and post-central gyri…did the patients all had injuries only in the lower limbs?
Line 216: The device is capable of stimulating five artificial neurons via surface electrodes that are placed on the skin at the pain site …. What does it mean? Reference 34 is not clear about mechanisms of action of ST.
Line 220: ST determine hypermetabolism in the cerebello-limbic system in patients with MCI following an electrical injury; it is quite different from the findings of the present work.
The discussion lacks a clear focus and hypothesis and must be improved.
Author Response
The paper could be of interest provided the authors report more precisely methods, results and data interpretation.
- Line 35: ….d severe chronic pain (a visual analogue scale [VAS] score of ≥5)… usually severe pain s for VAS >7/10; perhaps authors should indicate moderate-to-severe pain or moderate or severe pain; however, the most important issue is how disabling was the pain.
Answer> We appreciate you careful advise. The inclusion of pain was corrected to moderate or more pain, and a reference was added. We hope that these modifications will be better understood by the reader.
Jensen MP, Chen C, Brugger AM. Interpretation of visual analog scale ratings and change scores: a reanalysis of two clinical trials of postoperative pain. The journal of pain. 2003;4:407-14.
- Line 71: I am not sure that most readers are familiar with CBV techniques. The authors should spend 1 or 2 paragraphs explaining what it does actually measure and its advantages/disavantages compare to rsMRI and fMRI (Ke Zhang et al., 2018)
Answer> We appreciate you careful advise. As you recommended, we have added details about the advantages and disadvantages compared to other fMRIs and supplemented the references.
Park CH, Seo CH, Jung MH, Joo SY, Jang S, Lee HY, et al. Investigation of cognitive circuits using steady-state cerebral blood volume and diffusion tensor imaging in patients with mild cognitive impairment following electrical injury. Neuroradiology. 2017;59:915-21.
- Line 128: for patients with injuries on the left extremities, the map was flipped around the mid-sagittal axis…however, cortical non sensorimotor areas in the left and right hemisphere are not equivalent (Manss, 2019)...were patients all right-handed?
Answer> We agree with your point. We added that all participants were the dominant hand in the inclusion criteria. As the reviewer pointed out, the two hemispheres appear not to be symmetric in such a way that lateralized activation could be shown for some brain areas. We admit that CBV activation at the group level may have been affected by whether flipping of the brain was applied or not for the subjects with inconsistent handedness. However, given the limited number of subjects in the study, considering the side of injuries as another factor could have reduced the statistical power in statistical inferences, while we assumed that the effects of intervention on CBV activation would be much larger than the influence of the side of injuries. Indeed, in many clinical imaging studies such as for patients with stroke, flipping of the brain appears to be often applied to make lesions to be located in the same side primarily due to the limited sample size. As a limitation of the study related to the small sample size, we clearly described the point as well in the discussion section. We hope that these modifications will be better understood by the reader.
- Line 159: HRDS, Hamilton depression rating scale, scores are very low for patients with chronic intense pain…they do not seem real… in any event did depression scores changed after treatment?
Answer> We appreciate you careful advise. In this study, the Hamilton depression rating scale was performed to exclude patients with depression, and participants are considered to have low scores on the HRDS.
- Line 164: sham stimulation did not cause any change in pain VAS… however, placebo effect usually accounts for 20 to 60% of the analgesic effect…see Thibaut et al., 2019 (within authors references)
Line 168: …There was no difference in baseline CBV before ST between two groups…There should have been some (Joo et al., 2021)… why they did not find any?
Answer> We appreciate you careful advise. Both groups were the first patients admitted to the Department of Rehabilitation Medicine suffering from chronic pain after burns, and there was difference between the two groups before treatment. It is judged that there was differences in the results of the reference literature that was compared with normal adults without burn injuries.
- Which was the effect of the sham stimulation or, if you will, the placebo stimulation?
Answer> We appreciate you careful advise. During this study, pain medications were not completely stopped in both groups, the medications were maintained at the same dose, and non-pharmacologic treatments such as physical therapy, occupational therapy, and garment therapy were maintained. It is expected that the sham stimulation group, which was undergoing non-surgical treatment for hypertrophic scars, also had a pain-reducing effect. We added these limitations to the discussion section.
- Line 203: ST increase CBV in the left upper pre- and post-central gyri…did the patients all had injuries only in the lower limbs?
Answer> We appreciate you careful advise. We added information on the sites of burn injury to Table 2 and also added that there was no statistical difference between the two groups.
- Line 216: The device is capable of stimulating five artificial neurons via surface electrodes that are placed on the skin at the pain site …. What does it mean? Reference 34 is not clear about mechanisms of action of ST.
Line 220: ST determine hypermetabolism in the cerebello-limbic system in patients with MCI following an electrical injury; it is quite different from the findings of the present work.
Answer> We appreciate you careful advise. We have revised the sentences to make it easier for readers to understand.
- The discussion lacks a clear focus and hypothesis and must be improved.
Answer> We appreciate you careful advise. We have revised the methods, results, and discussion sections so that readers can understand this article.

Round 2
Reviewer 1 Report
Overall, the language of this manuscript is too strong. Conclusions must be softened and some clarifications must be made.
1) Lines 46-48: “The clinical effect of ST on burn pain and the potential mechanism of the clinical effect of treatment for 2 weeks were confirmed. ST may be a useful treatment option for patients who experience severe chronic pain related to burns.”
This can be simplified to the following: “ST may demonstrate an alternative strategy to manage chronic pain in burn patients.”
2) Line 58: “et el.”
The authors need to either write the examples or not include them. The reader has no way of knowing the treatments “et al” refers to.
3) Line 78: “chronic pain caused by burns”
Was this neuropathic or nociceptive pain? Or both?
4) Line 88-90: “moderate or more chronic pain (≥5 points on a 10-point visual analog scale [VAS]) lasting for >3 months after the burn injury, despite receiving pharmaceutical treatment and/or physical therapy[22, 23],”
Was this nociceptive pain or neuropathic pain? The pathophysiology for each is very different. How were these types of pain differentiated or diagnosed in these patients? This is important to know the target population to use this treatment.
5) Line 140-150:
“Statistical Analysis
All statistical analyses were performed using SPSS software (version 23.0; IBM Corp., Armonk, NY, USA). The non-parametric measurements between the two groups were analyzed using the Mann-Whitney test. To compare pretreatment homogeneity between groups, a Mann-Whitney U test was used for total body surface area (TBSA), days between burn injury and MRI acquisition, VAS score, BPI score, and HRDS score after normality test. Fisher’s exact test was used to assess sex and the sites of burn injury differences beween the groups. Pain scores from before and after treatment within each group were compared using the Wilcoxon signed rank sum test. Comparison of VAS scores of the two groups after 2 weeks treatment was compared using the Mann-Whitney test. Group differences and VAS correlations with a P-value of <0.05, after correction for multiple comparisons with threshold-free cluster enhancement, were considered statistically significant [32].”
If authors are using non-parametric measurements, results need to be reported as medians with interquartile ranges or minimums and maximums throughout the manuscript.
6) Lines 220-221: “For chronic pain after burns, the pain-reducing effect of ST and potential mechanisms of treatment related to hypersensitization were confirmed.”
Similarly, this is an overstatement. Consider “ST may demonstrate an alternative strategy to manage chronic pain in burn patients.”
7) Lines 258-259: “This is the first study to explore the effects of ST based on subjective and objective parameters, including cerebral activity in patients with chronic pain after burn injury.”
Authors can remove this sentence. It is difficult to know if this is truly the first. A patient presenting with miosis, ptosis, anhydrosis, and enophthalmos findings would be diagnosed as having “Horner syndrome,” now understood to represent an injury to the cervical sympathetic chain. If there is an associated brachial plexus palsy, then nerve root disruption and a poor spontaneous recovery are expected. Horner described this symptom complex in 1869. Silas Weir Mitchell, on pages 39 to 44 of his text in “Introduction l’tude de la mdecine exprimentale” (Paris, France; 1865), clearly describe a patient with this in 1864. In reality, this should be named “Mitchell syndrome.” They can start this paragraph with the limitations.
8) Line 259: “This study requires cautious of the data,”
This is not a complete sentence. Consider writing this as a different sentence. “This study presented many limitations.”
9) Line 276: “confirmed.” With the limitations this study presented, it is too strong of wording to say “confirmed.”
10) 274-276: “The clinical effect of ST on burn pain and the potential mechanism of the clinical effect of treatment for 2 weeks were confirmed. ST should be considered an effective alternative treatment of chronic pain in burn patients.”
This is an overstatement. This needs to be removed or revised with the many limitations. At most authors may write, “ST may demonstrate an alternative strategy to manage chronic pain in burn patients.”
Author Response
1) Lines 46-48: “The clinical effect of ST on burn pain and the potential mechanism of the clinical effect of treatment for 2 weeks were confirmed. ST may be a useful treatment option for patients who experience severe chronic pain related to burns.” This can be simplified to the following: “ST may demonstrate an alternative strategy to manage chronic pain in burn patients.”
Answer> We agree with your opinions. We modified the sentence without exaggerating the conclusions.
2) Line 58: “et el.”
The authors need to either write the examples or not include them. The reader has no way of knowing the treatments “et al” refers to.
Answer> We agree with your opinions. We removed “et el” from the sentence to avoid confusing the readers.
3) Line 78: “chronic pain caused by burns”. Was this neuropathic or nociceptive pain? Or both?
Answer> We appreciate you careful advise. In this study, neuropathic pain was described in indications throughout this article as a study on changes in the cerebral network due to chronic pain.
4) Line 88-90: “moderate or more chronic pain (≥5 points on a 10-point visual analog scale [VAS]) lasting for >3 months after the burn injury, despite receiving pharmaceutical treatment and/or physical therapy[22, 23],”. Was this nociceptive pain or neuropathic pain? The pathophysiology for each is very different. How were these types of pain differentiated or diagnosed in these patients? This is important to know the target population to use this treatment.
Answer> We appreciate you careful advise. The participants of this study were chronic neuropathic pain patients with skin epithelialization after acute burn treatment. It was stated that the participants were included for patients with reduced sensation due to skin sensory nerve damage caused by burns.
5) Line 140-150:
“Statistical Analysis
All statistical analyses were performed using SPSS software (version 23.0; IBM Corp., Armonk, NY, USA). The non-parametric measurements between the two groups were analyzed using the Mann-Whitney test. To compare pretreatment homogeneity between groups, a Mann-Whitney U test was used for total body surface area (TBSA), days between burn injury and MRI acquisition, VAS score, BPI score, and HRDS score after normality test. Fisher’s exact test was used to assess sex and the sites of burn injury differences beween the groups. Pain scores from before and after treatment within each group were compared using the Wilcoxon signed rank sum test. Comparison of VAS scores of the two groups after 2 weeks treatment was compared using the Mann-Whitney test. Group differences and VAS correlations with a P-value of <0.05, after correction for multiple comparisons with threshold-free cluster enhancement, were considered statistically significant [32].”If authors are using non-parametric measurements, results need to be reported as medians with interquartile ranges or minimums and maximums throughout the manuscript.
Answer> We appreciate you careful advise. We modified the table 1 and 2 as pointed out. We hope that these modifications will be well understood by readers.
6) Lines 220-221: “For chronic pain after burns, the pain-reducing effect of ST and potential mechanisms of treatment related to hypersensitization were confirmed.” Similarly, this is an overstatement. Consider “ST may demonstrate an alternative strategy to manage chronic pain in burn patients.”
Answer> We agree with your opinions. We modified the sentence without exaggerating the conclusions.
7) Lines 258-259: “This is the first study to explore the effects of ST based on subjective and objective parameters, including cerebral activity in patients with chronic pain after burn injury.”Authors can remove this sentence. It is difficult to know if this is truly the first.
Answer> We agree with your opinions. We removed the word of “first”.
A patient presenting with miosis, ptosis, anhydrosis, and enophthalmos findings would be diagnosed as having “Horner syndrome,” now understood to represent an injury to the cervical sympathetic chain. If there is an associated brachial plexus palsy, then nerve root disruption and a poor spontaneous recovery are expected. Horner described this symptom complex in 1869. Silas Weir Mitchell, on pages 39 to 44 of his text in “Introduction l’tude de la mdecine exprimentale” (Paris, France; 1865), clearly describe a patient with this in 1864. In reality, this should be named “Mitchell syndrome.” They can start this paragraph with the limitations.
Answer> This sentence does not appear to be relevant to this article.
8) Line 259: “This study requires cautious of the data,”
This is not a complete sentence. Consider writing this as a different sentence. “This study presented many limitations.”
Answer> We agree with your opinions. We modified the sentence as pointed out. conclusions.
9) Line 276: “confirmed.” With the limitations this study presented, it is too strong of wording to say “confirmed.”
274-276: “The clinical effect of ST on burn pain and the potential mechanism of the clinical effect of treatment for 2 weeks were confirmed. ST should be considered an effective alternative treatment of chronic pain in burn patients.”
This is an overstatement. This needs to be removed or revised with the many limitations. At most authors may write, “ST may demonstrate an alternative strategy to manage chronic pain in burn patients.”
Answer> We agree with your opinions. We modified the sentence without exaggerating the conclusions.

Reviewer 2 Report
The authors addressed the questions raised by the reviewers in a sterotypical fashion. Some questions are not properly annswered.
Author Response
The authors addressed the questions raised by the reviewers in a sterotypical fashion. Some questions are not properly annswered.
Answer> Answer> We appreciate you careful advise. The information you pointed out last time was a great help for future research.